# New Insights into the Transcriptional Regulation of Genes Involved in the Nitrogen Use Efficiency under Potassium Chlorate in Rice (*Oryza sativa* L.)

**DOI:** 10.3390/ijms22042192

**Published:** 2021-02-22

**Authors:** Nkulu Rolly Kabange, So-Yeon Park, Ji-Yun Lee, Dongjin Shin, So-Myeong Lee, Youngho Kwon, Jin-Kyung Cha, Jun-Hyeon Cho, Dang Van Duyen, Jong-Min Ko, Jong-Hee Lee

**Affiliations:** 1Department of Southern Area Crop Science, National Institute of Crop Science, RDA, Miryang 50424, Korea; rollykabange@korea.kr (N.R.K.); f55261788@korea.kr (S.-Y.P.); minitia@korea.kr (J.-Y.L.); jacob1223@korea.kr (D.S.); olivetti90@korea.kr (S.-M.L.); kwon6344@korea.kr (Y.K.); jknzz5@korea.kr (J.-K.C.); hy4779@korea.kr (J.-H.C.); kojmin@korea.kr (J.-M.K.); 2Molecular Biology Department, Agricultural Genetic Institute, Hanoi 11917, Vietnam; dangvanduyen79@gmail.com

**Keywords:** nitrogen use efficiency, transcriptional regulation, nitrate reductase, nitrate transporter, glutamate synthase, potassium chlorate, rice

## Abstract

Potassium chlorate (KClO_3_) has been widely used to evaluate the divergence in nitrogen use efficiency (NUE) between *indica* and *japonica* rice subspecies. This study investigated the transcriptional regulation of major genes involved in the NUE in rice treated with KClO_3_, which acts as an inhibitor of the reducing activity of nitrate reductase (NR) in higher plants. A set of two KClO_3_ sensitive nitrate reductase (NR) and two nitrate transporter (NRT) introgression rice lines (BC2F7), carrying the *indica* alleles of NR or NRT, derived from a cross between Saeilmi (*japonica*, P1) and Milyang23 (*indica*, P2), were exposed to KClO_3_ at the seedling stage. The phenotypic responses were recorded 7 days after treatment, and samples for gene expression, physiological, and biochemical analyses were collected at 0 h (control) and 3 h after KClO_3_ application. The results revealed that Saeilmi (P1, *japonica*) and Milyang23 (P2, *indica*) showed distinctive phenotypic responses. In addition, the expression of *OsNR2* was differentially regulated between the roots, stem, and leaf tissues, and between introgression lines. When expressed in the roots, *OsNR2* was downregulated in all introgression lines. However, in the stem and leaves, *OsNR2* was upregulated in the NR introgression lines, but downregulation in the NRT introgression lines. In the same way, the expression patterns of *OsNIA1* and *OsNIA2* in the roots, stem, and leaves indicated a differential transcriptional regulation by KClO_3_, with *OsNIA2* prevailing over *OsNIA1* in the roots. Under the same conditions, the activity of NR was inhibited in the roots and differentially regulated in the stem and leaf tissues. Furthermore, the transcriptional divergence of *OsAMT1.3* and *OsAMT2.3*, *OsGLU1* and *OsGLU2,* between NR and NRT, coupled with the NR activity pattern in the roots, would indicate the prevalence of nitrate (NO_3_¯) transport over ammonium (NH_4_^+^) transport. Moreover, the induction of catalase (CAT) and polyphenol oxidase (PPO) enzyme activities in Saeilmi (P1, KClO_3_ resistant), and the decrease in Milyang23 (P2, KClO_3_ sensitive), coupled with the malondialdehyde (MDA) content, indicated the extent of the oxidative stress, and the induction of the adaptive response mechanism, tending to maintain a balanced reduction–oxidation state in response to KClO_3_. The changes in the chloroplast pigments and proline content propose these compounds as emerging biomarkers for assessing the overall plant health status. These results suggest that the inhibitory potential of KClO_3_ on the reduction activity of the nitrate reductase (NR), as well as that of the genes encoding the nitrate and ammonium transporters, and glutamate synthase are tissue-specific, which may differentially affect the transport and assimilation of nitrate or ammonium in rice.

## 1. Introduction

Nitrogen is an essential macronutrient that plays an important role in the growth and development of plants [1]. In soil, about 95–99% of the potentially available nitrogen exists in organic form (plant or animal residues), in the soil organic matter, and in living soil organisms [2]. Most of the nitrogen available to plants is in the inorganic form, such as ammonium (NH_4_^+^) and nitrate (NO_3_^−^) ions [3,4], also called mineral nitrogen [5], while a very small amount of organic compounds, such as urea, may be available to the plant. NH_4_^+^ ions have a high affinity for binding to the negatively charged soil cation exchange complex (CEC), and act much like other cations in the soil. In contrast, because soil carries negative charges, NO_3_^−^ ions do not bind to soil particles, but are dissolved in the soil water, or precipitate in the form of soluble salts under dry conditions.

The nitrogen present in the soil that can be used by plants has two major sources: nitrogen-containing minerals, and the vast storehouse of nitrogen in the atmosphere. The biological conversion of the atmospheric nitrogen gas (N_2_) to ammonia (NH_3_^+^) is exclusively performed by bacterial and archaeal species. It is said that biological nitrogen fixation is specific-dependent, and not limited to particular genera. Therefore, the detection of N_2_ fixation is perceived as a complex task. For instance, in many legumes, such as legume symbionts, the fixation of nitrogen is strictly symbiotic [6].

The primary step of nitrogen acquisition by roots is the active transport across the plasma membrane of root epidermal and cortical cells. In rice, several members of nitrate transporter gene families (NPFs) have been identified, including 80 NPFs (NRT1/PRTs: NRT1, low-affinity nitrate transporter; PRT, di/tripeptide transporter), 5 NRT2s, 2 NAR2s members, and ammonium (NH_4_^+^) transporters (AMTs), and are reported to be involved in nitrogen use efficiency (NUE) [7]. Of this number, a few have been cloned and characterized [8,9,10,11,12,13,14,15,16,17]. Nitrogen assimilation is the formation of organic nitrogen compounds like amino acids from inorganic nitrogen compounds present in the environment. Organisms like fungi, certain bacteria, and the majority of plants that cannot fix nitrogen gas (N_2_) depend on the ability to assimilate nitrate or ammonia for their needs. Other organisms, like animals, depend entirely on organic nitrogen from their food [18].

Plant growth, development, and productivity require the permanent availability of nutrients, and the plant nutrient needs increase with the growth stage. Nitrogen, a key macronutrient is required in the process, among others. However, external fluctuations in the supply of nitrogen to plants can affect its uptake, and lead to activation of various regulatory networks in order to optimize N uptake and utilization [19]. During these events, nitrate reductase (NR) and nitrite reductase (NiR) convert the exogenous nitrate to ammonium (NH_4_^+^). In the process, NH_4_^+^ is further assimilated by glutamine synthase and glutamate synthase into amino acids [18]. In addition, it was earlier reported that the activity of a large number of enzymes involved in nitrogen assimilation decreases under long-term abiotic stress conditions [20], while short-term stress increases others [21]. In the chloroplasts [22], glutamine synthase incorporates this ammonia as the amide group of glutamine, using glutamate as a substrate. Glutamate synthase (Fd-GOGAT and NADH-GOGAT) transfers the amide group onto a 2-oxoglutarate molecule producing two glutamates [23].

Nitrogen is also vital because it is the major component of chlorophyll molecules, the key compound by which plants use the energy from sunlight to yield sugars from water and carbon dioxide during photosynthesis.

In our recent study, a novel quantitative trait locus (QTL) for chlorate resistance was identified, and candidate genes were proposed to be associated with chlorate resistance in rice [24]. A study conducted by Gao and his colleagues [25] supported that the *indica* allele of nitrate reductase (*OsNR2*) is the major component for the nitrogen biological cycle, and confers a high nitrogen use efficiency (NUE) compared to the *japonica* allele [26]. In the same way, Duan et al. [27] indicated that the *indica* allele of NR or NRT confers a high NUE, and a mutation in the NRT1.1B resulted in an impaired NUE. One of the major challenges in crop production and plant biology research is how to maintain a balanced crop productivity and yield, while reducing the application of nitrogen fertilizers and utilization by plants.

Several reports have supported the use of potassium chlorate (KClO_3_) as an effective and reliable strategy to investigate the nitrogen metabolism in higher plants, through the monitoring of nitrate uptake, transport, and assimilation, and knowing that nitrate is the main source of nitrogen. Chlorate (ClO_3_^−^) is a substrate for the nitrate reductase (NR) enzyme that reduces ClO_3_^−^ to the toxic chlorite, leading to a quick degradation of NR [28,29,30]. In wheat [31], ClO_3_^−^ resistance was used as a means of investigating the effect of NR antisense gene.

Potassium chlorate has been shown to induce stress in plants [24,32,33]. Generally, when plants are exposed to an environmental stimulus, they generate reactive oxygen species (ROS) [34,35] and reactive nitrogen species (RNS) [36]. Over accumulation of ROS or RNS may lead to oxidative damage [37] and lipid peroxidation [38], among other effects.

Therefore, this study aimed at investigating the transcriptional regulation of previously characterized genes reported to play important roles in nitrogen uptake, transport, assimilation, and remobilization; herein referred to as nitrogen use efficiency in rice. For this purpose, a set of nitrate reductase (NR) and nitrate transporter (NRT) introgression rice lines (BC2F7) and their related parental lines were exposed to potassium chlorate (KClO_3_) at seedling stage, and the phenotypic responses were evaluated in different rice tissues. Additionally, changes in the physiological properties of the parental and NR or NRT introgression rice lines were assessed in response to KClO_3_. Furthermore, the activity of key antioxidant enzymes and the change in the activity of the nitrate reductase (NR), as well as the extent of cell membrane degradation, were assessed under the same conditions.

## 2. Results

### 2.1. Distinctive Phenotypic Response between Parental Lines, and Identification of Introgression Lines

Initially, a population of 420 rice lines was screened through genotyping in order to identify NR or NRT introgression rice lines using insertion/deletion (InDel) markers (OsNR-IND2194 for nitrate reductase, and OsNRT-M10-22 for nitrate transporter). The results revealed that 26 lines and 59 lines carried the *indica* alleles of NR and NRT, respectively (Appendix A). These introgression lines amplified the expected band size of 200 bp for the *indica* allele of nitrate reductase, and 165 bp specific to the *indica* allele of nitrate transporter, respectively, on chromosomes 2 and 10, as reported earlier [24]. The *indica* alleles of the NR and NRT were previously suggested to explain the differences in the nitrogen use efficiency between *indica* and *japonica* rice varieties. Based on this evidence, the selected NR and NRT introgression lines (BC2F7) were exposed to 0.05% KClO_3_ for 7 days. Of this number, the introgression rice lines that exhibited a high sensitivity soon after KClO_3_ was applied (Appendix A), were used for physiological, biochemical, and molecular analyses.

Our data indicate that about 76.9% and 62.7% of NR and NRT introgression rice lines, showed a reduction in shoot growth, respectively, in response to KClO_3_ (Figure 1A). Under the same conditions, about 46.2% and 61% of the NR and NRT introgression lines recorded a decrease in root growth, respectively (Figure 1B). In the same way, 96.2% of the NR introgression lines showed a highly sensitive phenotype under KClO_3_, while for the NRT introgression lines, about 79.7% were KClO_3_ sensitive. The parental lines, Saeilmi and Milyang23, showed distinctive KClO_3_ responses. An increase in shoot growth was observed in Saeilmi (about 7.1%) and Milyang23 (about 0.1%) (Figure 1A and Figure 2I,K). Saeilmi exhibited a significant increase in root growth (about 32%), but Milyang23 showed a significant decrease in root growth (17.6%) under the same conditions (Figure 1B and Figure 2J,K).

In addition, panels A and B of Figure 2 revealed a normal distribution of shoot growth for both NR and NRT introgression lines, while the root growth of the same rice lines showed a positive skewness (Figure 2C) and a normal distribution (Figure 2D). However, the recorded frequency distribution of the chlorate response indicated a positive skewness from both the NR and NRT introgression lines (Figure 2E,F). From another perspective, the data in panel G of Figure 2 groups Saeilmi (P1) and Nipponbare (typical *japonica*) in close clusters (right side of the figure) with regard to their similarity in phenotypic response towards KClO_3_ treatment, while Milyang23 (P2) and 93-11 (typical *indica*) were assigned in the same cluster (left side of the figure). Moreover, the results of the principal component analysis (PCA) supported that root growth inhibition (RI) and chlorate were negatively correlated, whereas shoot growth inhibition positively correlated with the chlorate response of the NR and NRT introgression rice lines (Figure 2H). Principal component 1 (PC1) and 2 (PC2) explained 40.6% and 32.5% of the proportion of variance of the observed phenotypes, respectively, resulting in a cumulative proportion of 73.1%.

### 2.2. Exogenous Application of Potassium Chlorate Differentially Controlled the Activity of Nitrate Reductase in a Tissue-Specific Dependent Manner

Chlorates have been reported to have an inhibitory effect on the reducing activity of the nitrate reductase of nitrate (NO_3_^−^) to nitrite (NO_2_^−^). As expected, the activity of nitrate reductase (NR) was significantly suppressed by potassium chlorate (KClO_3_) in the roots of Saeilmi (P1), Milyang23 (P2), and 93-11 (typical *indica*), as well as in both the NRT introgression lines (NRT39 and NRT383), while in NR24 and NR144, a significant increase and a non-significant change were observed, respectively (Figure 3A). In the stem, the NR activity was suppressed in Saeilmi and Milyang23 soon after KClO_3_ was supplied, but significantly induced in 93-11 and in all the NR and NRT introgression lines (Figure 3B). Whereas, in the leaf tissues, Saeilmi, Milyang23, and 93-11 showed a similar increasing pattern of NR activity, but NR and NRT introgression lines differentially activated the NR enzyme under the same conditions (Figure 3C).

Upon stress induction, plants initiate an intrinsic adaptive response mechanism to combat the stress. This includes the activation of various antioxidant systems, tending to lower the over accumulation of reactive oxygen species (ROS) and alleviate the oxidative stress. Our data show that catalase (CAT) activity significantly increased in Saeilmi (P1, KClO_3_ resistant), but decreased in Milyang23 (P2, KClO_3_ sensitive) and 93-11 (typical *indica* cultivar). However, no significant change in CAT activity was observed in all NR or NRT introgression rice lines (Figure 3D). Other enzymatic antioxidants that have been shown to be induced by environmental cues are peroxidase (POD) and polyphenol oxidase (PPO). Here, we observed that POD activity did not change significantly in all tested rice lines upon KClO_3_ treatment (Figure 3E); however, PPO activity significantly increased in the Saeilmi (P1) and decreased in Milyang23 (P2) and 93-11 lines, but increased in the NR and NRT introgression lines, except in NRT383 (Figure 3F).

### 2.3. Increased Chlorophyll and Carotenoids Content in Response to Potassium Chlorate

Under normal growth conditions, the chloroplast pigments, chlorophyll (Chl) and carotenoids have been shown to play a crucial role in the acquisition and supply of the energy required by plants to complete their life cycle [39]. Several studies have reported a change in the chlorophyll and carotenoids content in response to environmental stimuli [40]. Our findings show that the Chl *a* (Figure 4A), Chl *b* (Figure 4B), and total chlorophyll (Figure 5C), as well as the carotenoids relative to the chlorophyll (C*x+c*, Figure 4D), significantly increased in all the tested rice lines in response to KClO_3_ application. Likewise, the pheophytin (Figure 4E–G), known as a derived molecule of chlorophyll lacking the magnesium at the reaction center, as well as the carotenoids relative to the pheophytin (Cph *x+c*, Figure 4H), showed a similar accumulation pattern to that observed for the Chl, but much more abundant compared to Chl.

### 2.4. Potassium Chlorate Treatment Caused Lipid Peroxidation and Changes in Proline Content

Under stressful conditions, particularly caused by an environmental stimulus, various free radicals identified as reactive oxygen species (ROS) are generated. Over accumulation of ROS induces oxidative stress that may result in oxidative damage. In plants, malondialdehyde (MDA) is commonly used as a biomarker to estimate the lipid peroxidation due to damage to the cell membrane integrity. The results indicated that MDA content significantly decreased in the roots of Saeilmi (P1, chlorate resistant), while a significant increase was recorded in Milyang23 (P2, KClO_3_ sensitive), 93-11 (typical *indica*), and all the NR and NRT introgression rice lines in response to KClO_3_ (Figure 5A). A similar pattern of NR activity was observed in the stem (Figure 5B) and the leaf tissues (Figure 5C). Furthermore, proline has been reported to accumulate under abiotic stress in plants, as part of the adaptive response mechanism towards stress tolerance. Here, we observed a significant increase in the proline content in Saeilmi and NR24, as well as in NRT39 and NRT383, soon after KClO_3_ was applied (Figure 6D). Under the same conditions, 93-11 and NR144 showed a significant reduction in proline content.

### 2.5. Potassium Chlorate Differentially Regulated the Expression of Genes Involved in Nitrogen Uptake, Transport, and Assimilation in Roots, Stem, and Leaf Tissues

Four KClO_3_ sensitive NR (NR24, NR144) and NRT (NRT39 and NRT383) introgression rice lines and the parental lines, as well as 93-11 (the typical *indica* and KClO_3_ sensitive), were used to investigate the transcriptional regulation of key genes involved in nitrogen uptake, transport, and assimilation, herein referred to as nitrogen use efficiency (NUE), in different plant organs (roots, stem, and leaf) under potassium chlorate (KClO_3_) treatment. The relative expression levels of all analyzed genes were normalized to those of the housekeeping gene (*OsActin1*), and Milyang23 (P2, *indica*) was used to assess the statistical significance of the transcripts accumulation of the target genes in different rice lines. Our data showed that *OsNR2* was upregulated in the roots, stem, and leaf of Milyang23 (P2, about 1.5, 2.2, and 2.4-fold changes, respectively), while in Saeilmi (P1), a non-significant change was observed in the roots and leaf tissues (Figure 6A,C), but upregulated in the stem (2.2-fold change) (Figure 6B). When expressed in the NR introgression rice lines, the *OsNR2* was shown to be downregulated in the roots; while being upregulated in the stem and leaves. However, the transcripts accumulation of *OsNR2* was significantly decreased in all tested plant organs in both NRT39 and NRT383 rice lines, but was upregulated in 93-11 in the roots, and downregulated in the stem and leaves.

After being uptaken, nitrogen (N) is transported into the cell through the cell membrane in two major forms, nitrate (NO_3_^−^) and ammonium (NH_4_^+^). N transport in the form of nitrate, also known as the major form in which nitrogen is carried, is achieved through the combinational action of different enzymes, of which *OsNRT1.1B* has been suggested to play a preponderant role. Here, the expression of *OsNRT1.1B* was downregulated in the roots of Milyang23 (about 1-fold change), 93-11 (0.4-fold change), and NR24 (0.5-fold change), as well as in both NRT39 and NRT383 lines (0.4-fold change) (Figure 6D). A similar transcriptional pattern was observed when *OsNRT1.1B* was expressed in the stem, but in Saeilmi no significant change was recorded between the KClO_3_ treated and control plants, with an exception being in the leaf tissues, where an increase in the transcripts accumulation was recorded in Saeilmi (0.7-fold change upregulated), Milyang23 (1.6-fold change upregulated), and 93-11 (0.7-fold change), as well as in the NR and NRT introgression lines (Figure 6E,F). However, *OsNRT1.1B* expression was downregulated in the NRT39 and NRT383 lines.

N transport in the form of NH_4_^+^ is facilitated by a group of well-identified NH_4_^+^ transporters, including *OsAMT1.3* and *OsAMT2.3* genes. Our data show that the expression of *OsAMT1.3* was downregulated in the roots of Saeilmi (P1, about 1-fold change), while being upregulated in Milyang23 (P2, about a 1.4-fold change) and 93-11 (1.2-fold change). However, *OsAMT1.3* was upregulated in the NR (1.0 and 1.1-fold change) and downregulated in the NRT (0.4 and 0.5-fold change) introgression lines (Figure 6G). Similarly, when expressed in the stem, the data in panel H of Figure 3 indicates that *OsAMT1.3* was downregulated in Saeilmi (0.4-fold change) and 93-11 (0.6-fold change), while showing an increased transcript accumulation pattern in Milyang23 (2.4-fold change) and NR (1.3 and 1.6-fold change) introgression lines, but being downregulated in the NRT (0.6 and 0.4-fold change) introgression lines (Figure 6H). Meanwhile, *OsAMT1.3* transcripts levels decreased in the leaf tissues of Saeilmi (about 1-fold change), while showing an increased pattern in 93-11 (about 1-fold change), as well as in all NR (4.3 and 2.3-fold change) and NRT (about 1.0 and 1.7-fold change) introgression lines, except NRT39 (Figure 6I), in response to KClO_3_.

The other ammonium transporter of which the transcript accumulation was measured (*OsAMT2.3*) was shown to be induced by KClO_3_ in the roots of the Saeilmi (1.1-fold change), Milyang23 (1.3-fold change), and 93-11 (1.7-fold change) cultivars. However, an opposite expression pattern was recorded in the NR (about 1-fold downregulated and 1.2-fold upregulated) and NRT (0.6 and 0.5-fold change, downregulated) introgression lines (Figure 6J). Under the same conditions, the transcripts accumulation of *OsAMT2.3* in the stem increased in Saeilmi (2.1-fold change), Milyang23 (1.3-fold change), and 93-11 (1.7-fold change), as well as in both NR introgression lines (1.7 and 1.4-fold change), but a decrease was observed in the NRT (0.7 and 0.8-fold change) introgression lines (Figure 6K). When expressed in the leaf tissues, *OsAMT2.3* exhibited a similar transcript accumulation pattern to that observed in the stem (Figure 6L).

The N that is uptaken by the roots, and transported through the vessels into to the leaf tissues via the stem is expected to be assimilated in the form of amino acids. In higher plants, glutamate synthase has been reported as being involved in the initial steps of the N assimilation. Therefore, in order to investigate the possible effect of KClO_3_ on the N assimilation events, we monitored the transcript accumulation of glutamate synthase (*OsGLU1* and *OsGLU2*) genes. Our data indicate that *OsGLU1* was downregulated in the roots and stem of Saeilmi (P1) and Milyang23 (P2), as well as in NR24, NRT39, and NRT383 (Figure 6M,N). Whereas, a non-significant change in *OsGLU1* expression was observed (Figure 6O). In the same way, *OsGLU2* was downregulated in both the roots (Figure 6P) and the stem (Figure 6Q) of Saeilmi, but a slight increase was recorded in Milyang23 and 93-11, while a downregulation was recorded in the NR and NRT introgression lines. In the stem, *OsGLU2* transcript accumulation increased in 93-11 and decreased in Milyang23, as well as in all NR and NRT introgression lines (Figure 6P,Q). Meanwhile, when expressed in the leaves, the expression of *OsGLU2* was upregulated in Saeilmi and in the NR introgression lines, while being downregulated in Milyang23, 93-11, and the NRT introgression lines (Figure 6R).

As part of the nitrogen (N) biological cycle, particularly during the denitrification process, where nitrate (NO_3_^−^) is reduced back to nitrous oxide (N_2_O) and nitrogen (N_2_), nitric oxide (NO) is released. For this reason, we investigated the changes in the transcripts accumulation of key NO biosynthetic genes, *OsNIA1* and *OsNIA2*, encoding a nitrate reductase enzyme, in response to KClO_3_. Our findings revealed, on the one hand, that *OsNIA1* was upregulated in the roots of Saeilmi (1.2-fold change), Milyang23 (1.4-fold change), 93-11 (1.2-fold change), NR24, and NR144 (1.6 and 1.3-fold change, respectively), but downregulated by 0.7 and 0.5-fold change in NRT39 and NRT383, respectively (Figure 6S). Similarly, a significant upregulation of *OsNIA1* was recorded in the stems of both Saeilmi (P1) and Milyang23 (P2) (1.9 and 2.2-fold change, respectively), and the NR and NRT introgression lines (NR: 5.2 and 1.3-fold change; NRT: 1.1 and 1.2-fold change) (Figure 6T). However, *OsNIA1* transcript accumulation decreased in the leaf tissues of both P1 and P2 (0.4 and 0.3-fold change, respectively), and 93-11 (0.3-fold change), while being upregulated by about a 1-fold change in all the NR and NRT introgression lines (Figure 6U). On the other hand, *OsNIA2* transcript accumulation increased significantly in the roots of Saeilmi (3.2-fold change), and was much higher in Milyang23 (22.3-fold change) and in the NR (2.8 and 3.9-fold change) and NRT (424.8 and 6948.3-fold change) introgression lines (Figure 6V). The expression of *OsNIA2* was upregulated in the stem of Saeilmi (1.0-fold change), Milyang23 (1.9-fold change), but downregulated in 93-11 (0.3-fold change), as well as in NR (0.5 and 0.8-fold change) and NRT (0.8 and 0.7-fold change). Likewise, *OsNIA2* expression showed a similar downregulation pattern in the leaf tissues of all tested rice lines (Figure 6W,X).

## 3. Discussion

### 3.1. Potassium Chlorate Inhibits Shoot and Root Growth in a Cultivar Dependent Manner

Roots are important plant organs playing an essential role during plant nutrients and water uptake, as well as for anchoring the plant to the soil. Roots also serve as storage organs supporting the whole plants structure and fitness, among other things [41]. In the same way, the shoots (above ground part of the plant, including the stem or tillers and the leaves and grains) are important for the plant architecture, transport of nutrients and water, energy acquisition, and redistribution, and carry the target products of many food crops, such as rice. Nitrate (NO_3_^−^) and ammonium (NH_4_^+^) are widely known as the major forms of nitrogen (N) utilized by plants from the environment (soil) [42]. Under stressful conditions caused by environmental stimuli [43,44,45], roots are the first organs that sense the stress, and the availability of essential nutrients, including nitrogen (N), is affected, leading to impairment of root growth and/or shoot growth and productivity. Here, we recorded that about 46.2% and 61% of the NR or NRT introgression rice lines exhibited a reduction in their roots, and 76.9% and 62.7% showed a reduction in shoot growth, respectively, in response to KClO_3_, suggesting a strong inhibitory effect of KClO_3_ on the whole plant growth.

### 3.2. Potassium Chlorate Differentially Regulates Genes Involved in Nitrate Uptake, Transport, and Assimilation in a Tissue-Specific Dependent Manner

The *indica* allele of the nitrate reductase encoded by the *OsNR2* gene (chromosome 2) and the nitrate transport, *OsNRT1.1B* (chromosome 10) have been suggested to be key players in explaining the differences in the nitrate assimilation capacity and the nitrogen use efficiency (NUE) between *indica* and *japonica* rice subspecies [25]. According to the authors, this difference is conferred by allelic variation at the *OsNR2*, with the *indica* allele showing a high nitrate reductase (NR) activity compared to its *japonica* counterpart. Consequently, the *indica* allele was suggested to promote NO_3_^−^ uptake, while interacting with the nitrate transporter encoding gene, *OsNRT1.1B*. However, NR activity has been reported to be inhibited by chlorates (ClO_3_^−^), which together with NO_3_^−^ serve as substrates for the NR enzyme that reduces NO_3_^−^ to the toxic chlorite [28,29,30,46]. Many studies have used ClO_3_^−^ to isolate mutant plants that are defective in nitrate reduction [29], while others have suggested that when ClO_3_^−^ is applied to plants, an increase in NR mRNA level is observed, but not the NR protein content [47]. Zhao and his colleagues [48] observed that a chlorate resistant mutant (lacking the NR activity) of rice exposed to ClO_3_^−^ showed a similar level of NR activity to the wild type.

Here, the tissue-specific expression of the *OsNR2* gene in rice lines carrying the *indica* alleles of NR or NRT revealed a differential transcriptional regulation by KClO_3_. When expressed in the roots, the expression of *OsNR2* was inhibited in all introgression lines, except in NRT144 line (Figure 6A), but its expression in the stem (Figure 6B) and the leaves (Figure 6C) was shown to be significantly upregulated in the NR introgression lines, while being downregulated in all the NRT introgression lines. Therefore, these results suggest that the previously reported inhibitory effect of ClO_3_^−^ on the reducing activity of the nitrate reductase (NR) is tissue-specific, rather than systematic to all plant tissues and organs. This is also supported by the differential nitrate reductase activity, which was shown to vary between rice plants’ tissues (roots, stem, and leaf tissues) (Figure 6A–C). In addition, the contrasting expression pattern of the *OsNR2* gene in the NRT introgression lines compared to that recorded in the NR introgression background suggests an antagonistic relationship, particularly in the stem and the leaf tissues. Furthermore, the differential transcriptional regulation of *OsNR2* in different plant tissues would imply that NR may also be involved in NO_3_^−^ long-distance signaling and transport across the plant body.

Another set of genes known to code for nitrate reductase enzymes in plants are *NIA1* and *NIA2*. In rice, *OsNIA1* and *OsNIA2* have been shown to be induced by nitric oxide (NO), one of the compounds that is generated during the nitrogen metabolic process [49]. In *Arabidopsis*, mutations in the nitrate reductase structural genes *AtNIA1* and *AtNIA2* have been shown to result in an impairment in NO_3_^−^ assimilation rate [50]. It could then be said that the recorded significant increase in the transcript accumulation of *OsNIA1* in the NR introgression rice lines background in all measured rice tissues, and the downregulation in the NRT introgression rice lines background in response to KClO_3_ (Figure 6S–U), coupled with the upregulation or downregulation of *OsNIA2* expression in a tissue-dependent manner (Figure 6V–X), would suggest a possible role of NO in nitrogen use efficiency.

In the same way, *OsNRT1.1B,* known to function in NO_3_^−^ uptake, transport, and signaling [51], exhibited a differential expression between the parental lines Saeilmi and Milyang23, as well as in different tissues of the NR or NRT introgression lines. Furthermore, the transcript accumulation of *OsNRT1.1B* was significantly lower in the NRT, compared to the NR, introgression line upon KClO_3_ treatment, which would suggest that the NR enzyme prevails over the NRT in NO_3_^−^ uptake as part of the NUE. From another perspective, *OsNRT1.1B* has been shown to be involved in the regulation of root microbiota to facilitate organic nitrogen mineralization in the soil, while mediating plant–microbe interactions [52]. *OsNRT1.1B* was also reported to transport N under both low and high NO_3_¯ levels, and had an essential single nucleotide polymorphism (SNP) between *indica* and *japonica* rice subspecies [53], where the *indica* variation was shown to contribute to the NUE divergence between rice *indica* and *japonica*, eventually by promoting NO_3_^−^ uptake and translocation, and upregulation of NO_3_^−^ response [17].

Likewise, nitrogen is also transported in the form of ammonium (NH_4_^+^). In the soil, NH_4_^+^ basically results from the mineralization of organic matter, and represents, besides NO_3_^−^, the quantitatively most important source of N for plant nutrition [54]. The authors supported that despite the low concentrations in soils, the uptake of NH_4_^+^ by the plant can be achieved at a very high rate, facilitated by the multiple transporters in the root plasma membrane. A recent report [55] suggested that, in addition to functioning in NH_4_^+^ uptake in the roots, *OsAMT1.3* may act as a signal sensor to regulate plant growth, carbon, and nitrogen use efficiency (NUE). Similarly, *OsAMT1.3* has been suggested as being involved in the adaptation ability of rice to low NH_4_^+^ supplies [56]. In the same way, *OsAMT2.3* was reported to be affected by N supply [57]. Therefore, the recorded upregulation of *OsAMT1.3* in the roots, stem, and leaves of NR introgression lines, and the downregulation in NRT introgression lines (Figure 6G–I) soon after KClO_3_ was applied, coupled with the transcriptional patterns of *OsAMT2.3* under the same conditions (Figure 6J–L), would indicate an antagonistic relationship with NR, while suggesting a co-expression with NRT, in rice.

Glutamate synthase (GOGAT) is involved in the initial steps of nitrogen assimilation in plants. The GS/GOGAT cycle is widely known as the major route of NH_4_^+^ assimilation in higher plants [58]. Hence, the recorded transcriptional levels of Glutamate synthase 1 and 2 encoding genes (Figure 6M–O and Figure 2P–R) would imply that *OsGLU2* would prevail over *OsGLU1* in N assimilation.

### 3.3. Potassium Chlorate Differentially Regulates Antioxidant Enzymes Between Parental and NR or NRT Introgression Rice Lines

Generally, when plants are exposed to a changing environment, various ROS are generated [59,60]. To cope with the stress, plants activate an array of antioxidant systems as part of their adaptive response mechanism, which include catalase (CAT), peroxidase (POD), and polyphenol oxidase (PPO), in order to maintain a low level of ROS accumulation, and keep a balanced reduction–oxidation state within the cell [61,62]. However, over accumulation of ROS was shown to induce oxidative stress, which may result in oxidative damage, and may culminate in the induction of programmed cell death (PCD) [63,64]. Owing to the recorded increase in CAT and PPO activities in the *japonica* parental line (Saeilmi, chlorate resistant) and the decrease in the *indica* parental line (Milyang23, chlorate sensitive), coupled with the differential activity in the NR or NRT introgression lines, this study suggests that CAT and PPO play an important role in the initial adaptive response mechanism towards KClO_3_ resistance.

During oxidative stress, plants accumulate ROS, which exacerbates oxidative damage [37]. In the process, malondialdehyde (MDA), commonly used as a marker to estimate lipid peroxidation in plant species, accumulates abundantly to the extent of the degradation of the cell membrane [38]. Therefore, the recorded MDA accumulation patterns in different plant organs (roots, stem, and leaf tissues) (Figure 5A–C) indicate the level of oxidative stress caused upon KClO_3_ application, suggesting that the integrity of the cell membrane might have been affected as a result of oxidative damage.

### 3.4. Exogenous Application of Potassium Chlorate Triggers the Accumulation of Chloroplast Pigments

Chlorates (ClO_3_^−^) were earlier reported as potent inhibitors of the reducing activity of NR in plants, which may have a significant effect on N metabolism. As is well known, assimilation of the inorganic form of nitrogen (N) is as of fundamental importance to crop growth, as it is to crop productivity. It is said that N is the main plant mineral nutrient required for the production of chlorophyll (Chl), as well as other components of the plant’s cells, such as proteins, nucleic acids, and amino acids [51]. As ClO_3_¯ is said to inhibit NR activity, which may impair NUE, and owing to the fact that N is essential for chlorophyll production, we were expecting to see a reduction in the chlorophyll content upon KClO_3_ application. Rather, we recorded a significant increase in the chlorophyll *a* and *b* contents and their immediate precursors and degradation products, pheophytin [65] and carotenoids, in all tested rice lines (parental lines and BC2F7 NR or NRT introgression lines) (Figure 4A–H). In our recent studies, chlorophyll content was shown to be increased in response to abiotic stresses, such as drought stress [62] and salinity [66]. It is then thought that the observed Chl and pheophytin accumulation, as well as the carotenoids content, were a result of the oxidative stress caused by the application of KClO_3_, which suggests these molecules as emerging biomarkers for assessing the overall plant health status under oxidative stress conditions, and in which proline would contribute as an osmoprotectant [67,68].

Therefore, from the perspective of providing a comprehensive view of the recorded morphological, physiological, and biochemical changes, as well as the molecular responses of the nitrate reductase (NR) or nitrate transporter (NRT) introgression rice lines (BC2F7) relative to their parental lines Saeilmi (P1, *japonica*) and Milyang23 (P2, *indica*), moving towards potassium chlorate treatment, in the roots, stem and the leaf tissues, a signaling model is proposed (Figure 7).

## 4. Materials and Methods

### 4.1. Plant Materials and Potassium Chlorate Treatment

To perform the experiments, a total of seven rice lines, including two parental lines (Saeilmi: *japonica* and Milyang23, *indica*), the *indica* reference cultivar 93-11, and four introgression lines carrying the *indica* alleles of nitrate reductase (NR, two lines) or nitrate transporter (NRT, two lines) derived from a cross between Saeilmi and Milyang23, were used as genetic materials. Initially, cv. Nipponbare (typical *japonica* ssp.) and cv. 93-11 (typical *indica* ssp.) were used to identify an optimum potassium chlorate (KClO_3_) concentration. For this purpose, the two cultivars were subjected to a gradient of concentrations of KClO_3_ (0.01%, 0.025%, 0.05%, 0.07%, and 0.1%) for seven days at seedling stage. The control treatments were supplied with distilled water only. Based on the phenotypic responses, 0.05% KClO_3_ was selected as the concentration to be used for downstream analyses. At 0.05% KClO_3_, a significant difference of the shoot (Appendix A) or root (Appendix A) inhibition percentage between Nipponbare and 93-11 was obtained. Meanwhile, the gap in the biomass fresh weight (BFW) between Nipponbare and 93-11 was small at 0.05% KClO_3_ level (Appendix A). Therefore, 0.05% was used for downstream experiments.

Prior to germination, rice seeds were sterilized with 0.7% nitric acid (HNO_3_) (CAS: 7697-37-2, Lot No. 2016B3902; Junsei Chemical Co. Ltd., Tokyo, Japan) overnight to break the dormancy [24], followed by incubation for 48 h at 27 °C to induce germination. Germinated seeds were grown in 50-well trays containing an enriched soil, and placed in a greenhouse until three-leaf stage. Seedlings with a uniform height (3-week old, six seedlings per treatment per rice line) were transferred into 50 mL falcon tubes prior to KClO_3_ treatment. Then, the roots of seedlings were immerged into 5–10 mL of 0.05% potassium chlorate (KClO_3_) (CAS: 3811-04-9, Lot No. BCBW5513; Sigma-Aldrich, St. Louis, MO, USA) solution (pH 5.6), and placed in a growth chamber under dark conditions for 7 days at ±25 °C [33]. The KClO_3_ solution was replaced three days after the initial application by irrigation method. The control seedlings were supplemented with distilled water only. The phenotypic response was recorded 7 days after KClO_3_ application. The chlorate resistance was calculated as the percentage of the ((total number of tested seedlings − number of dead seedlings in KClO_3_)/total number of tested seedlings) × 100. However, shoot inhibition was estimated as the percentage of the ((shoot length under control (SLC)—shoot length under KClO_3_ (SL_KClO_3_)/SLC) × 100. The same formula was used to estimate the roots inhibition percentage. Samples (leaf, stem, and roots) for gene expression, biochemical, and physiological analyses were collected in triplicate at 0 h (untreated control) and 3 h after treatment, and immediately frozen in liquid nitrogen and kept in a −80 °C freezer for further processing.

### 4.2. Genomic DNA Extraction, Genotyping, and Molecular Marker Analysis

The genomic DNA was extracted from leaf samples using the previously described CTAB method with slight modifications [40]. Briefly, frozen leaf samples were crushed in 1.5 mL Eppendorf tubes (e-tubes). Then, 600 μL of 2× CTAB buffer (D2026, Lot D2618U12K; Biosesang, Seongnam-si, Korea) was added, and the mixture was vortexed and incubated for 30 min at 65 °C in a dry oven. A solution containing 500 μL of PCI (Phenol:Chloroform:Isoamylalcohol, 25:24:1, Batch No. 0888k0774; Sigma-Aldrich, St. Louis, MO, USA) was added, followed by gentle mixing by inversion. The tubes were centrifuged for 15 min at 13,000 rpm, and the supernatant was transferred to fresh e-tubes, followed by the addition of 500 μL of isopropanol (CAS: 67-63-0, Lot No. SHBC3600V; Sigma-Aldrich, St. Louis, MO, USA), mixing by inversion, incubation at −20 °C for 1 h, and centrifugation at 13,000 rpm for 7 min. The supernatant was removed and the pellets were washed with 70% ethanol (1 mL). Samples were centrifuged at 13,000 rpm for 2 min and ethanol was discarded, followed by drying at room temperature and re-suspension in 100 μL 1× TE buffer (Lot No. 0000278325; Promega, Madison, WI, USA).

A population consisting of 420 rice lines was genotyped in order to identify introgression lines carrying the *indica* alleles of the nitrate reductase (NR) or nitrate transporter (NRT) genes through polymerase chain reaction (PCR) using OsNR-IND2194 and OsNRT-M10-22 insertion/deletion (InDel) markers. The reaction mixture (15 µL) consisted of 1.5 µL 10X reaction buffer, 0.8 µL 10 mM dNTP, 1 µL 10 pM primers (forward and reverse), 0.1 µL *Taq* polymerase, and adjusted to the final volume with nuclease-free water. A 3-step cycling reaction was performed including polymerase activation at 95 °C for 5 min, strand separation at 94 °C for 20 s, annealing at 56–59 °C for 30 s for 35 cycles, extension at 72 °C for 1 min/kb, and a final extension at 72 °C for 5 min. The amplicons were separated on 3% agarose gel electrophoresis, and the bands were visualized using a gel documentation system. The sequences of InDel primers can be found in Appendix A.

### 4.3. Total RNA Extraction, cDNA Synthesis, and qPCR Analysis

Total RNA was extracted from leaf samples using the TaKaRa MiniBEST Universal Plant RNA Extraction Kit (TAKARA Bio Inc., Cat. No. 9769 v201309Da, Kusatsu, Japan) following the manufacturer’s instructions. Briefly, frozen samples with liquid nitrogen were ground to fine powder, and 450 μL buffer RL (containing 50× dithiothreitol (DTT, 20 μL per 1 mL buffer RL) was added, and the mixture was pipetted up and down for few seconds until the lysate showed no precipitate, followed by centrifugation at 12,000 rpm for 5 min at 4 °C. The supernatant was transferred to fresh 1.5 mL Eppendorf tubes (e-tubes) (step 1). Then, 1 volume of ethanol (100%) was added to the mixture from step 1, followed by pipetting up and down to mix well, and 600 μL were transferred to a spin column with a 2 mL collection tube. The tubes were centrifuged at 12,000 rpm for 1 min, and the flow-through was discarded. Then, 500 μL buffer RWA was added, followed by centrifugation at 12,000 rpm for 30 sec, and the flow-through was discarded. Next, 600 μL buffer RWB was added to the spin column, followed by centrifugation at 12,000 rpm for 30 sec (this step was repeated twice), and the flow-through was discarded. Then, empty spin columns with collection tubes were centrifuges for 2 min at 12,000 rpm, and the spin columns were placed on fresh 1.5 mL e-tubes, followed by the addition of 100 μL RNase free water. Samples were incubated at room temperature for 5 min, followed by centrifugation at 12,000 rpm for 2 min (the elution step was repeated twice).

For cDNA (complementary DNA) synthesis [69], 1 µg of RNA was used as a template and the ProtoScript^®^ II First Strand cDNA Synthesis Kit (New England BioLabs Inc., NEB Labs, MA, USA) was employed, as described by the manufacturer. The cDNA was then used as a template for qPCR (quantitative polymerase chain reaction) to investigate the transcript accumulation of the selected genes (see Appendix A).

For gene expression analysis, a reaction mixture comprising 10 μL Prime Q-Master Mix (with SYBR green I), 0.1× ROX (0.1 μL/50×) (GENETBIO Inc., Daejeon, Korea) along with 1 μL of template DNA and 10 pM of each forward and reverse primer in a final volume of 20 µL reaction. A no-template control (NTC) was used. A 3-step reaction including polymerase activation at 95 °C for 10 min, a denaturation at 95 °C for 20 s, annealing at 60 °C for 30 sec and extension at 72 °C for 30 sec was performed in a Real-time PCR machine (QuantStudio™ Design and Analysis Software v.1.3, Applied Biosystems, Thermo Fisher Scientific, Seoul, Korea). Total reaction cycles were 40 and the data were normalized with relative expression of rice Actin1. The list of the genes with their corresponding primers sequences used in the study is given in Appendix A.

### 4.4. Proline Measurement Assay

Proline quantification was done following the colorimetric method described earlier [70]. Approximately 100 mg leaf samples were homogenized in 3% sulfosalicylic acid (5 µL × mg^−1^ FW) in Eppendorf tubes (e-tube). Then, the homogenate was centrifuged using a benchtop centrifuge at 13,000 rpm for 5 min. Then, 100 µL from the supernatant of the plant extract was added to the reaction mixture (100 µL of 3% sulfosalicylic acid, 200 µL glacial acetic acid, 200 µL acidic ninhydrin (1.25 g ninhydrin (1,2,3-indantrione monohydrate), 30 mL glacial acetic acid, 20 mL of 6 M orthophosphoric acid (H_3_PO_4_)), dissolved into double distilled water and stored at 4 °C). The mixture was incubated at 96 °C for 1 h and immediately cooled on ice to terminate the reaction. The samples were extracted with toluene by adding 1 mL toluene to the reaction mixture and vortexing for about 20 s, and then leave for 5 min on the bench in order to allow separation of the organic and water phases. Then the chromophore (upper phase colored light red) containing toluene was moved into the fresh cuvette and the absorbance was read at 520 nm using toluene as reference. The proline concentration was calculated on the fresh weight basis and expressed in µg g^−1^ FW.

### 4.5. Catalase, Polyphenol Oxidase, and Peroxidase Activity Assay

The activity of antioxidant enzymes was assayed using the spectrophotometric method as described by Elavarthi and Martin [71]. Briefly, 100 mg of leaves were ground to a fine powder using liquid nitrogen (N_2_) and immediately homogenized in 50 mM phosphate buffer (pH = 7.5). The mixture was centrifuged for 10 min at 12,000 rpm at 4 °C after being kept on ice for about 10 min. The supernatant was then transferred to fresh 1.5 mL e-tubes. The homogenate was used as a crude enzyme source for catalase (CAT), peroxidase (POD), and polyphenol oxidase (PPO) activities.

Catalase activity was assayed as described earlier [49]. Briefly, a volume of 50 µL H_2_O_2_ (50 m*M*) was added to the crude enzyme extract, and the absorbance of the reaction mixture was measured at 240 nm wavelength after one minute. The activity of catalase was expressed in units per milligram of sample fresh weight [72].

The spectrophotometric method was also used to estimate the activity of peroxidase (POD) and polyphenol oxidase (PPO), as earlier described by Khan*,* et al. [73]. The supernatant obtained after centrifugation in the previous paragraph was used as crude enzyme source. For POD, the reaction contained 50 µL crude extract, 50 µL pyrogallol (50 µM), 25 µL H_2_O_2_ (50 mM), and 10 µL phosphate buffer (0.1 M), and incubated for 5 min in dark conditions. Then, 25 µL H_2_SO_4_ (5% *v*/*v*) was added to stop the reaction. The absorbance was measured at 420 nm wavelength [74]. For PPO activity analysis, the reaction mixture was composed of 50 μL pyrogallol (50 mM) and 100 μL phosphate buffer (0.1 M). The absorbance of the reaction was read at 420 nm wavelength, and calculations were done as previously described [75].

### 4.6. Chlorophyll, Pheophytin, and Carotenoids Content Measurements

The photosynthesis process has been shown to be crucial for plant growth and development, and productivity. Chlorophyll, pheophytin, and total carotenoids content were measured as described by Lichtenthaler [76]. About 200 mg leaf tissues were homogenized with acetone (80%), followed by centrifugation for 10 min at 3000 rpm. The supernatant was transferred to fresh falcon tubes, and the centrifugation was repeated until all chlorophyll was harvested in the solvent. The combined supernatant was made up to a known volume with 80% acetone. The absorbance of the extract was read at 645, 663, and 652 nm for chlorophyll *a*, *b*, and total chlorophyll; 480 and 510 nm for total carotenoids; and 665, 653, and 470 nm for pheophytin *a*, *b*, and total pheophytin. Acetone was used as a blank. The OD values at 645, 663, and 652 were used to calculate chlorophyll contents and total carotenoids as described earlier [77].

### 4.7. Lipid Peroxidation Assay

The lipid peroxidation level in the leaf tissue mentioned here was measured as the malondialdehyde (MDA) content, determined by thiobarbituric acid (TBARS) reaction [78]. About 200 mg of non-treated and treated samples were homogenized in 4 mL Trichloroacetic acid (TCA) (0.1%) with a porcelain mortar and pestle, followed by centrifugation for 15 min at 10,000 rpm. Then, 1 mL of the supernatant was harvested and 2 mL of TCA (20%) containing 0.5% TBA was added. The supernatant was incubated in a preheated water bath at 95 °C for 30 min and immediately cooled on ice. The reaction mixture was centrifuged for 10 min at 10,000 rpm, and the optical density (OD) was measured at 532 nm and 600 nm wavelength (OD_600_ as the non-specific absorbance that is subtracted from the OD_532_ reading). MDA level was calculated using Lambert’s equation (extinction coefficient of MDA 155 nM^−1^ × cm^−1^).

### 4.8. Nitrate Reductase Activity Assay

The nitrate reductase activity was assayed following an in vivo spectrophotometric method, as previously described [79], with slight modifications. NR enzyme activity was determined by measuring the amount of nitrite (NO_2_^−^) released from plant tissues. Briefly, about 2 g of samples were homogenized into 2 mL reaction buffer composed of 40 m*M* potassium nitrate (KNO_3_), 0.08 M Sodium phosphate dibasic (Na_2_HPO_4_), 0.02 *M* Sodium phosphate monobasic (NaH_2_PO_4_), and 4% (*v*/*v*) n-propanol (CH_3_CH_2_CH_2_OH), with a pH of 7.5. The mixture was centrifuged at 12,000 rpm for 10 min. The supernatant was incubated for 2 h in dark conditions at room temperature, and the reaction was stopped by adding of 200 µL of 1% sulfanilamide (H_2_NC_6_H_4_SO_2_NH_2_) dissolved in 3N HCl and 200 µL of 0.05% *N-*(1-naphthyl) ethylenediamine hydrochloride (C_10_H_7_NHCH_2_CH_2_NH_2_ · 2HCl). We, therefore, determined NO_2_^−^ concentration by reading the absorbance of the solution at *A_540_* nm. In the case the absorbance was greater than 0.5, the reaction solution was diluted 10-fold with the reaction buffer and sulfanilamide.

### 4.9. Statistical Analysis

All the experiments were performed using a completely randomized design (CRD). The data were collected in triplicates and analyzed statistically with GraphPad Prism software (Version 7.00, 1992–2016 GraphPad, San Diego, CA, USA). Analysis of variance (ANOVA) for CRD was performed, and the Turkey’s multiple comparison was employed at a significance level of 0.05.

## 5. Conclusions

Understanding nitrogen (N) metabolism in higher plants and all aspects involved in the process is essential to improving nitrogen use efficiency (NUE) and rationalizing N application to plants. Here, a set of potassium chlorate (KClO_3_) sensitive rice lines (BC2F7) carrying the *indica* alleles of the nitrate reductase (NR) or the nitrate transporter (NRT), were exposed to KClO_3_ at seedling stage. The parental lines recorded distinctive phenotypic responses 7 days after treatment. In addition, *OsNR2* expression was differentially regulated between the roots, stem, and leaf tissues, and between introgression lines. Similarly, the expression pattern of *OsNIA1* and *OsNIA2* in the roots, stem, and leaves indicated that they are differentially regulated by KClO_3_. Furthermore, the transcriptional divergence of the ammonium transporters and that of the glutamate synthase encoding genes and associated with the pattern of NR activity in the roots, would indicate the prevalence of nitrate (NO_3_^−^) transport over ammonium (NH_4_^+^) transport. Moreover, the increase or decrease of catalase (CAT) and polyphenol oxidase (PPO) enzyme activities, coupled with the changes in the chloroplast pigments and proline contents, as well as the accumulation of malondialdehyde (MDA) revealed the extent of the oxidative damage by KClO_3_, resulting in lipid peroxidation. All results suggest that the inhibitory effect of KClO_3_ on the reducing activity of the nitrate reductase (NR), as well as that of the genes encoding the ammonium transporters and glutamate synthase are tissue-specific, rather than systematic to all plant tissues or organs in rice.

## Figures and Tables

**Figure 1 ijms-22-02192-f001:**
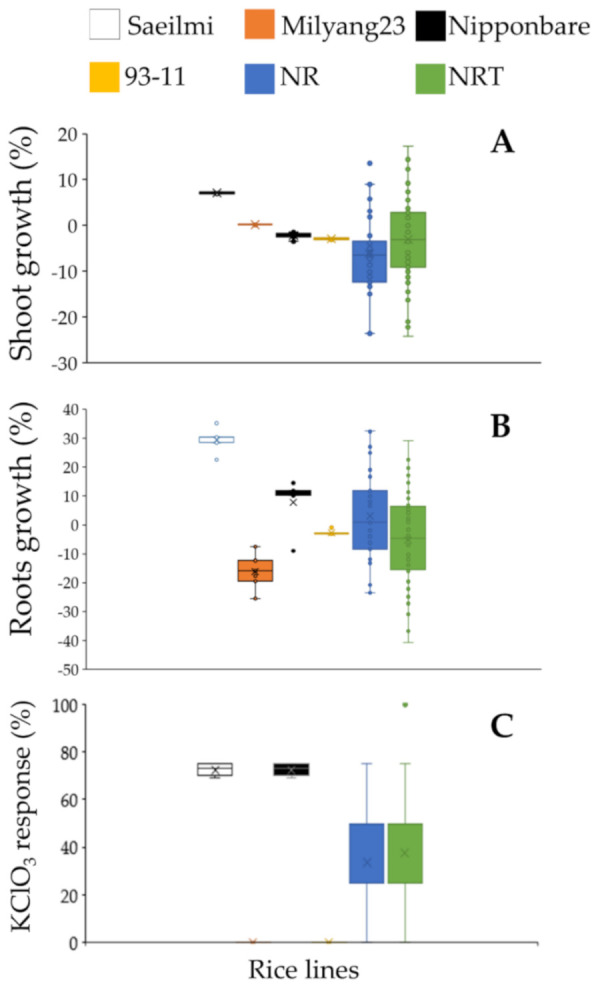
Genotype–phenotype correlation of the nitrate reductase and nitrate transporter introgression lines and parental line in response to potassium chlorate (KClO_3_). (**A**) The box plots display the shoot growth patterns of the Saeilmi (P1, *japonica*), Milyang23 (P2, *indica*), Nipponbare (typical *japonica* cultivar), 93-11 (typical *indica* cultivar), nitrate reductase (NR, *n* = 26), and nitrate transporter (NRT, *n* = 59) introgression lines. (**B**) Box plots showing the root growth patterns, and (**C**) the KClO_3_ response.

**Figure 2 ijms-22-02192-f002:**
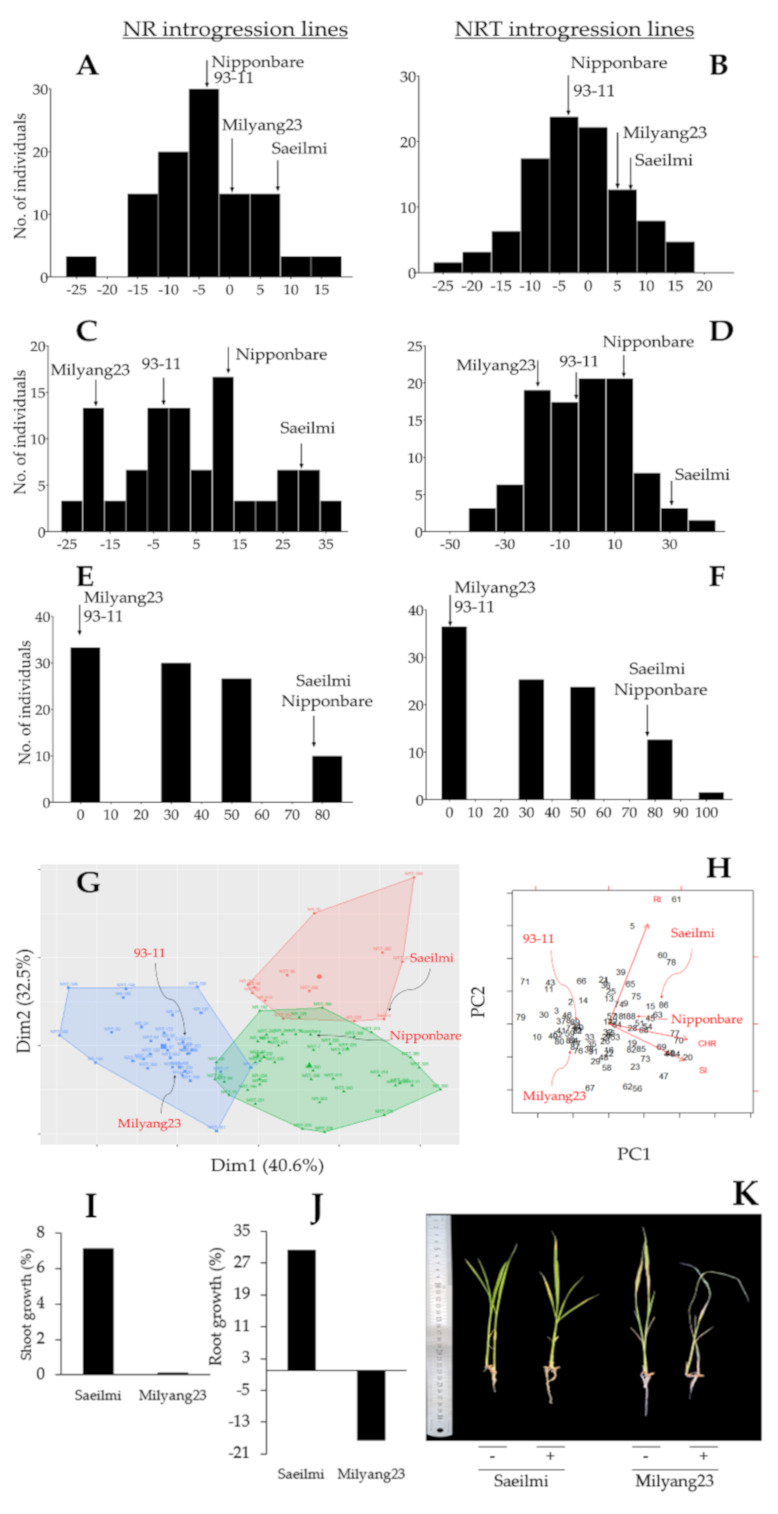
Frequency distribution, clustering, and principal component analysis results. Frequency distribution of the shoot growth of the nitrate reductase (NR, *n* = 26) (**A**), and nitrate transporter (NRT, *n* = 59) (**B**) introgression rice lines showing a normal distribution, (**C**) frequency distribution of the roots growth of NR, and (**D**) NRT introgression lines showing a positive skewness and a normal distribution, respectively. (**E**) frequency distribution of the KClO_3_ response of NR, (**F**) NRT introgression rice lines showing a positive skewness, (**G**) Clusters showing the distinctive phenotypic response between Saeilmi (P1) and Milyang23 (P2), and that of the NR or NRT introgression lines, and (**H**) 2-D principal component analysis (PCA) indicating the correlation between traits, and (**I**–**K**) shoot and root growth pattern of Saeilmi (P1) and Milyang23 (P2) in response to KClO_3_.

**Figure 3 ijms-22-02192-f003:**
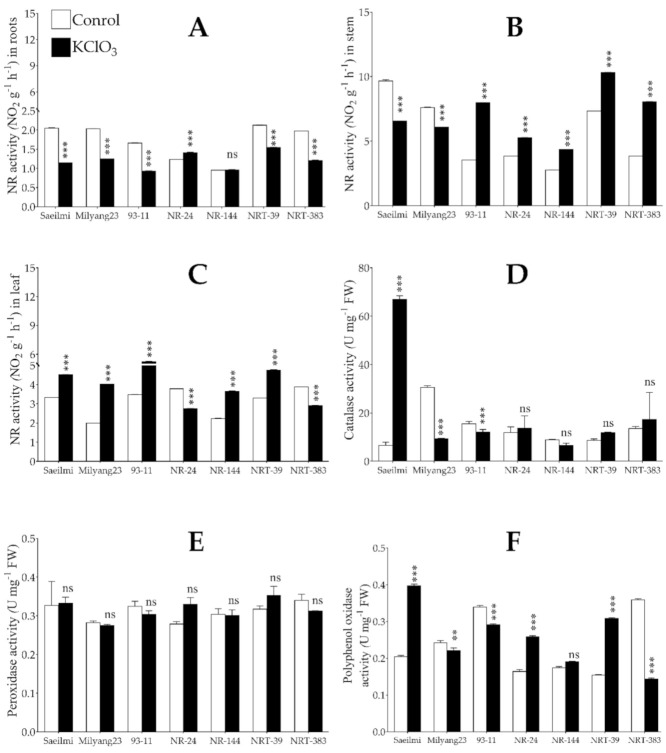
Changes in the activity of nitrate reductase (NR) and antioxidant enzymes. (**A**–**C**) Tissue-specific activity of the nitrate reductase (NR) enzyme in response to 0.5% potassium chlorate (KClO_3_) 3 h after treatment, (**D**) catalase (CAT) activity, (**E**) peroxidase (POD) activity, and (**F**) polyphenol oxidase (PPO) activity in roots of rice seedlings exposed to KClO_3_. White bars are controls, while black bars are KClO_3_ treated seedlings. Bars are mean values of triplicates ± SE. **** p* < 0.001, *** p* < 0.01, *ns* non-significant.

**Figure 4 ijms-22-02192-f004:**
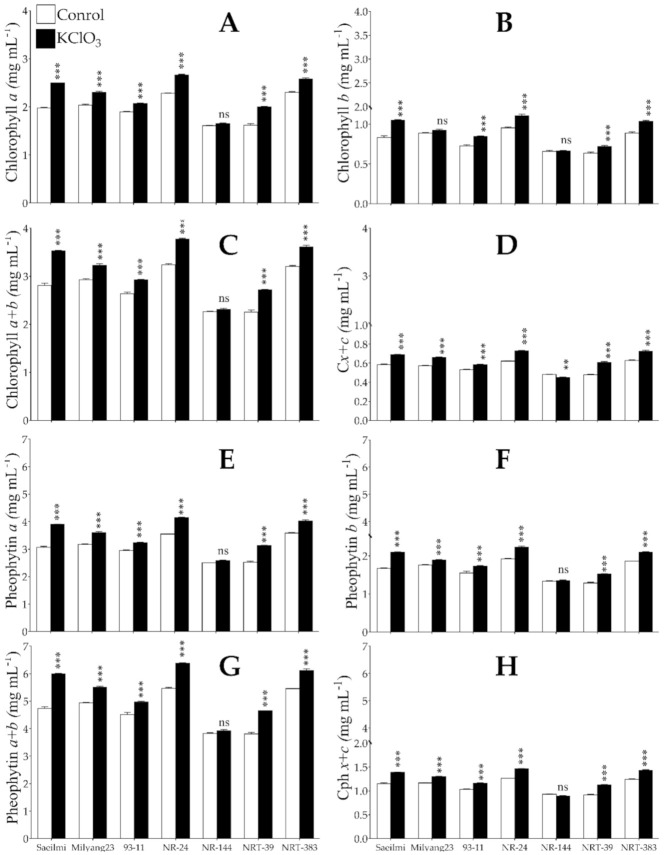
Pattern of chloroplast pigment content under potassium chlorate treatment. (**A**–**D**) Chlorophyll *a*, *b*, *a*+*b*, and carotenoids relative to chlorophyll (C*x+c*), respectively, (**E**–**H**) Pheophytin *a*, *b*, *a*+*b*, and carotenoid relative to pheophytin (C*ph x+c*), respectively. Bars are mean values of triplicates ±SE. **** p* < 0.001, *** p* < 0.01, *ns* non-significant.

**Figure 5 ijms-22-02192-f005:**
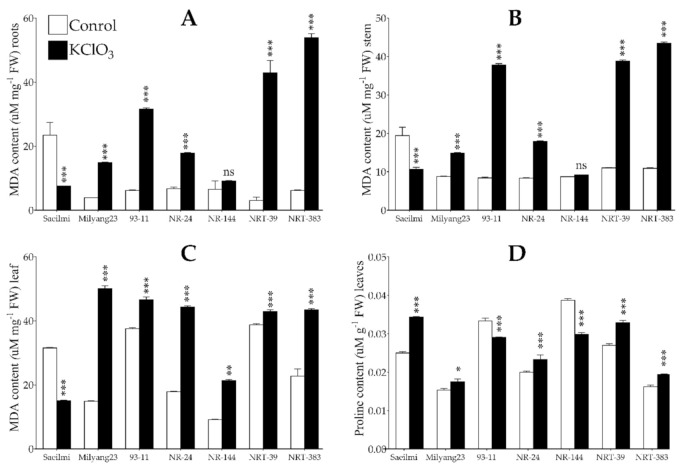
Lipid peroxidation and proline change in proline content. (**A**) Lipid peroxidation by malondialdehyde (MDA) content in the roots, (**B**) stem, and (**C**) leaf tissues in response to potassium chlorate (KClO_3_) treatment, and (**D**) pattern of proline accumulation under KClO_3_ treatment. Bars are mean values of triplicates ±SE. **** p* < 0.001, *** p* < 0.01, ** p* < 0.05, *ns* non-significant.

**Figure 6 ijms-22-02192-f006:**
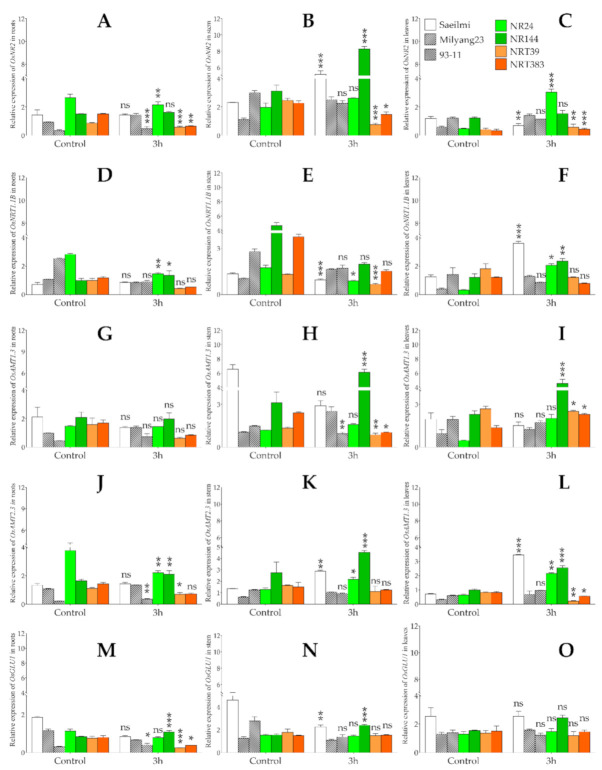
Transcriptional patterns of major nitrogen use efficiency (NUE) genes in different rice tissues under potassium chlorate (KClO_3_) treatment. Transcriptional patterns of major NUE genes in different rice tissues under potassium chlorate treatment. (**A**–**C**) Transcriptional pattern of *OsNR2* in the roots, stem, and leaves of three week-old nitrate reductase (NR) and nitrate transporter (NRT) introgression rice lines in response to 0.05% potassium chlorate (KClO_3_), soon after treatment. (**D**–**F**) Transcript accumulation of *OsNRT1.1B*, (**G**–**I**) *OsAMT1.3*, (**J**–**L**) *OsAMT2.3*, (**M**–**O**) *OsGLU1*, (**P**–**R**) *OsGLU2*, (**S**–**U**) *OsNIA1*, and (**V**–**X**) *OsNIA2* under the same conditions. Bars are mean values of triplicates ±SE. **** p* < 0.001, *** p* < 0.01, ** p* < 0.05, *ns* non-significant. Data were compared with the expression level of Milyang23 (P2, KClO_3_ sensitive) for the statistical significance.

**Figure 7 ijms-22-02192-f007:**
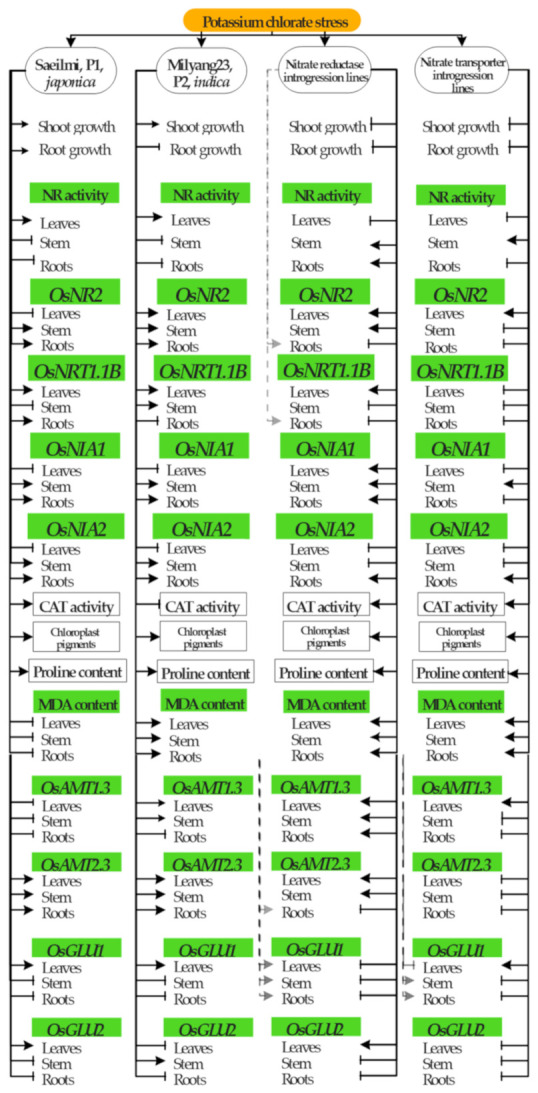
Signaling model summarizing the morphological, physio-biochemical, and molecular response of NR and NRT introgression rice lines towards potassium chlorate. The phenotypic responses, the physiological and biochemical changes, as well as the molecular responses of the parental lines Saeilmi (P1, *japonica*) and Milyang23 (P2, *indica*) and their derived introgression lines (nitrate reductase or nitrate transporter, BC2F7) are summarized in the above signaling model. Continuous lines with an arrow indicate upregulation (for gene expression) or induction/increase (for enzyme activity or accumulation of physiological components). Continuous lines with a perpendicular bar indicate downregulation or inhibition/decrease. Dotted lines with or without an arrow or a perpendicular bar indicate that one of the introgression lines differentially expressed a particular gene in the corresponding rice tissue. This signaling model was created using ConceptDraw Pro v.10.3.2.114 (CS Odessa Corp, San Jose, CA, USA).

## Data Availability

The data presented in this study are available in supplementary material.

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
