# Peer review of "New Insights into the Transcriptional Regulation of Genes Involved in the Nitrogen Use Efficiency under Potassium Chlorate in Rice (Oryza sativa L.)"

_ijms, 2021, doi:10.3390/ijms22042192_

Round 1

Reviewer 1 Report

The manuscript by Kabange et al., is of great interest for the effects of potassium chlorate on NR, genes encoding the nitrate and ammonium transporters, and glutamate synthase. The experiments were elaborately planned and well executed. The results were well described. I have minor suggestions that could strengthen the manuscript.

  1. In 2.1 “Distinctive Phenotypic Response Between Parental, and Identification of Introgression Lines”, the authors should show pictures of different phenotypes along with figure 2.
  2. The authors stated that inhibitory potential of KClO3 on the reducing activity of the NR is tissue-specific tissue specific. Have the authors also measure concentration of KClO3 in different tissues to see if the altered activity of NR is directly associated with KClO3 concentrations in those tissues.
  3. The application of KClO3 in these experiments was done under normal growth conditions, Have the authors try the condition of nitrogen depravation?

Author Response

New Insights into the Transcriptional Regulation of Genes Involved in the Nitrogen Use Efficiency under Potassium Chlorate in Rice (Oryza sativa L.)

Manuscript ID: ijms-1108523

Point by point response to the comments of reviewers

We are thankful to the editorial team and anonymous reviewers for their time given to this manuscript. We appreciate their comments, and are happy to share that most of the comments are addressed and have substantially improved the quality of the manuscript. We would like to specify that all changes in the manuscript were highlighted green, no track change was applied to the manuscript. We hope that the manuscript in the present form will be suitable for publication in the journal.

Reviewer 1

The manuscript by Kabange et al., is of great interest for the effects of potassium chlorate on NR, genes encoding the nitrate and ammonium transporters, and glutamate synthase. The experiments were elaborately planned and well executed. The results were well described. I have minor suggestions that could strengthen the manuscript.

1. In 2.1 “Distinctive Phenotypic Response Between Parental, and Identification of Introgression Lines”, the authors should show pictures of different phenotypes along with figure 2.

We are thankful to the reviewer for his concern. We have included the phenotype of the parental lines Saeilmi (P1, japonica) and Milyang23 (P2, indica) as suggested.

2. The authors stated that inhibitory potential of KClO3 on the reducing activity of the NR is tissue-specific tissue specific. Have the authors also measure concentration of KClO3 in different tissues to see if the altered activity of NR is directly associated with KClO3 concentrations in those tissues.

The authors appreciate this concern raise by the reviewer. We have observed that the inhibitory potential of KClO3 on the reducing activity of the NR is tissue-specific. In this study, we did not quantify the KClO3 in different plant tissues. But, we compared the activity of nitrate reductase (NR) and nitrate transporter (NRT), coupled with the transcripts accumulation of the genes encoding these enzymes, in KClO3 treated rice seedlings compared to those grown without treatment, building on the fact that chlorates are the major inhibitors of NR activity. In addition, the potassium chlorate and other ions uptake by the plants and their transport across the plants organs as described earlier [1], may require specific and independent experiment that might include gradient concentrations. From another perspective, Jiemei and his colleagues [2] reported that KClO3 inhibited nitrate reductase in the leaves, but increased nitrogen concentration as well as the amino acids content in the leaves and buds of Dimocarplus longan Lour Trees, therefore suggesting the activation of de novo synthesis of amino acids.

3. The application of KClO3 in these experiments was done under normal growth conditions, Have the authors try the condition of nitrogen depravation?

Authors appreciate the concern raised by the reviewer. We would like to indicate that after the seedlings were transferred from the trays to the falcon tubes, KClO3 (prepared with distilled water only) was applied, and the control seedlings were grown in water (distilled water) without any nutrient.

  1. Keltjens, W. Factors affecting absorption and transport of potassium in maize roots; Pudoc: 1978.
  2. Jiemei, L.; Ruitao, Y.; Huicong, W.; HUANG, X. Stress Effects of Chlorate on Longan (Dimocarpus longan Lour.) Trees: Changes in Nitrogen and Carbon Nutrition. Horticultural Plant Journal 2017, 3, 237-246.

Reviewer 2 Report

I have read the manuscript entitled:” New insights into the transcriptional regulation of genes involved in the nitrogen use efficiency under potassium chlorate in rice (Oryza sativa L.)” by Kabange et al. In this research article the authors used several rice lines, including a cross providing introgression lines, to assess the role of two sets of major gene loci in the NUE under potassium chlorate treatment. They employed gene targeted transcriptional and biochemical approaches in different rice tissues. The manuscript is well written and the findings interesting. However, some modification in the presentation of the results could be improved, also some more information in the method section would help strengthening the results.

Minor comments:

Line 66: “various” could be replaced by further detailed information and numbers in order to give a better idea to the reader. Similarly, the notion of “major genes” line 111 should be clearly defined.

Line 94: add “for” after “component.

Line 103. Capital letter to add to “Chlorate”

Line 112: add “nitrogen” after “the”

Line 207: add “in” after downregulated

Line 251: Meaning is not clear to me

Line 285: “T” in “tested”

In the methods, it would be appreciated if the authors could indicate if the tissues used were fresh, frozen, or other for the various processing done.

Line 580: remove “d”

Line 653: is something missing in this sentence?

References should be carefully checked as several are incomplete, eg no journal name line 711

Major comments:

The authors should consider moving the gene expression data at the end, ie after the phenotypic and biochemical data, this (I think) would help the reader to better appreciate the difference identified between lines based on their phenotyping responses.

The authors should consider a schematic to summarize the main significant findings.

In the result section, I would suggest the addition of fold change levels for some relevant comparison to help strengthening the point of the authors and link more easily the transcriptional and biochemical changes when significant. Line 255 “slightly increased”, line 256 “significantly increased” and line 261 “much higher” are a bit confusing, please consider the significance and add some fold changes to avoid meaningless wording.

Differences between introgression lines for the same loci should be discussed.

Figure 3: The choice of Milyang23 at 3h to compare for the gene expression data (line 249) should be further explained/justified at the beginning of the corresponding result section.

qPCR analysis: The authors should provide the PCR efficiencies for the primer pairs and the range of expression levels for the Actin 1 in the different tissues and line used, so that the variations of the targeted genes are not due to the variation in the reference gene.

Author Response

New Insights into the Transcriptional Regulation of Genes Involved in the Nitrogen Use Efficiency under Potassium Chlorate in Rice (Oryza sativa L.)

Manuscript ID: ijms-1108523

Point by point response to the comments of reviewers

We are thankful to the editorial team and anonymous reviewers for their time given to this manuscript. We appreciate their comments, and are happy to share that most of the comments are addressed and have substantially improved the quality of the manuscript. We would like to specify that all changes in the manuscript were highlighted green, no track change was applied to the manuscript. We hope that the manuscript in the present form will be suitable for publication in the journal.

Reviewer 2

I have read the manuscript entitled:” New insights into the transcriptional regulation of genes involved in the nitrogen use efficiency under potassium chlorate in rice (Oryza sativa L.)” by Kabange et al. In this research article the authors used several rice lines, including a cross providing introgression lines, to assess the role of two sets of major gene loci in the NUE under potassium chlorate treatment. They employed gene targeted transcriptional and biochemical approaches in different rice tissues. The manuscript is well written and the findings interesting. However, some modification in the presentation of the results could be improved, also some more information in the method section would help strengthening the results.

Minor comments:

Line 66: “various” could be replaced by further detailed information and numbers in order to give a better idea to the reader.

Similarly, the notion of “major genes” line 111 should be clearly defined.

We are thankful to the reviewer for the suggestion. We have improved the statement as suggested (lines 66-70).

Likewise, the sentence in lines 114-115 (previously on line 111) has been rephrased

Line 94: add “for” after “component.

Line 96 (previously line 94): we added “for” after component and before the nitrogen…

Line 103. Capital letter to add to “Chlorate”

Line 106 (previously line 103): we capitalized the initial “C” of chlorate as suggested.

Line 112: add “nitrogen” after “the”

Line 117, previously line 113, we have added “nitrogen” as suggested.

Line 207: add “in” after downregulated

Line 292 (previously line 207): we have included as suggested.

Line 251: Meaning is not clear to me

Lines 340-361, we have rephrased the statement as follows: As part of the nitrogen (N) biological cycle, particularly during the denitrification process, where nitrate (NO3) is reduced back to nitrous oxide (NO2) and nitrogen (N2), nitric oxide (NO) is released.

Line 285: “T” in “tested”

Line 291 (previously 285), we corrected the typo mistake. “rested” was change with “tested”.

In the methods, it would be appreciated if the authors could indicate if the tissues used were fresh, frozen, or other for the various processing done.

We are thankful to the reviewer for the concern. We have included a statement indicating the following in line 506-508: Samples (leaf, stem, roots) for gene expression, biochemical and physiological analyses were collected in triplicate at 0 h (untreated control) and 3 h after treatment, and immediately frozen in liquid nitrogen and kept in –80 °C for further processing.

Line 580: remove “d”

Line 587 (previously line 580), we removed the “d” as suggested.

Line 653: is something missing in this sentence?

We are thankful to the reviewer for the suggestion to improve the statement in the conclusion. Therefore, we have modified the statement as suggested (lines 699-700)

References should be carefully checked as several are incomplete, eg no journal name line 711

We strongly appreciate the concern of the reviewer to improve the quality of the manuscript. We have included the missing information and removed the duplicated citations from the manuscript and the list of references as recommended.

Major comments:

The authors should consider moving the gene expression data at the end, ie after the phenotypic and biochemical data, this (I think) would help the reader to better appreciate the difference identified between lines based on their phenotyping responses.

We appreciate the suggestion made by the reviewer to reorganize the result section. We have no objection with the suggestion. Therefore, we have moved the gene expression data and modified the order of the figures accordingly.

The authors should consider a schematic to summarize the main significant findings.

The authors appreciate the suggestion made by the reviewer, and express no objection to that. Therefore, we have proposed a signaling model summarizing the findings of the study (see Figure 7) (lines 502-513)

In the result section, I would suggest the addition of fold change levels for some relevant comparison to help strengthening the point of the authors and link more easily the transcriptional and biochemical changes when significant.

Line 255 “slightly increased”, line 256 “significantly increased” and line 261 “much higher” are a bit confusing, please consider the significance and add some fold changes to avoid meaningless wording.

We are thankful to the reviewer for the comments that helped us improved the manuscript. We have included the change in the expression level in fold change as suggested (lines 258-324) and lines 338-359.

Differences between introgression lines for the same loci should be discussed.

Figure 3: The choice of Milyang23 at 3h to compare for the gene expression data (line 249) should be further explained/justified at the beginning of the corresponding result section.

Figure 6 (previously figure 3): Milyang23 is the P2, one of the parental lines carrying the indica alleles of the NR and NRT genes, which exhibited a high chlorate sensitivity. The indica alleles of nitrate reductase and nitrate transporter have been reported to explain the divergence in the nitrogen use efficiency between japonica and indica rice cultivars. Therefore, we have included a justification in lines 260-263.

qPCR analysis: The authors should provide the PCR efficiencies for the primer pairs and the range of expression levels for the Actin 1 in the different tissues and line used, so that the variations of the targeted genes are not due to the variation in the reference gene.

The authors appreciate the concern raised by the reviewer. However, we would like to specify the following: Generally, the relative expression of any gene by qPCR is normalized to that of the housekeeping gene (i.e. Actin or Ubiquitin), which implies that the resulted expression of the target genes in different samples (control and treated) has been already normalized during the analysis (using Actin as the reference gene). Then, the differences between the control and treated samples are evaluated statistically.

Round 2

Reviewer 2 Report

The authors have significantly improved the manuscript and addressed most of my comments. As a minor comment I would strongly suggest that the authors present the PCR efficiencies of the primer pairs and corresponding amplicon length so that their qRTPCR data are strengthened to further support their conclusions.

Author Response

The authors would like to thank the reviewer for the concern. We have included the required information in the supplementary table (Table S1) as shown below:

Table S1. List of primer sequences for gene expression used in the study

Marker/gene names

Locus

Forward primer (5’->3’)

Reverse primer (5’->3’)

Length

(F/R)

Tm °C

(F/R)

GC contents (F/R)

Amplicon size (bp)

Genotyping InDel marker primers

OsNR-IND2194

LOC_Os02g53130

/BGIOSGA005531

GTGCTGACCTCACGTCCATC

GTAGCCCGAGCTTCTGGTC

20/ 19

59.5/ 59.2

60/63.2

200 (indica)/ 188 (japonica)

OsNRT-M10-22

LOC_Os10g40600/ BGIOSGA031434

TCGCGTGACAAATATGACAT

CCACTGCAAGATCCAAGTCT

20/ 20

51.3/ 55.4

50/45

165 (indica)/ 213 (japonica)

High affinity nitrate transporters encoding gene

OsNRT1.1B

LOC_Os10g40600

GGCTCGACTACTTCTACTGGC

CGAGGCGCTTCTCCTTGTAG

21/20

59.8/59.5

57.1 / 60

102

High affinity ammonium transporters encoding genes

OsAMT1.3

LOC_Os02g40710

GCGCGCTCTTCTACTACCTC

GTAGTCGTACCCTGTCTGCG

20/ 20

59.5/ 59.5

60/ 60

120

OsAMT2.3

LOC_Os01g61550

CGGATGAACATCAAGGCGTG

TATCCGCCGGAGTAGTCGAT

20/ 20

57.5/ 57.5

55.0/ 55

116

Glutamate synthase involved in the initial step of nitrogen assimilation

GLU1

LOC_Os01g48960

TGTTGCTGTCAGTTCGCTCT

AAACCCAACAAGGGTGCAGA

20/20

55.4/ 55.4

50.0/ 50.0

140

GLU2

LOC_Os05g48200

AATGCTCTTCCCAACCCTGG

CTGCTGTTAATCCGTGCTGC

20/20

57.5/ 57.5

55.0/ 55.0

110

Nitrate reductase catalyzing the conversion of nitrate (NO3) to nitrite (NO2)

OsNR2

LOC_Os02g53130

TCCTCGCCTACATGCAGAAC

ATGCGCTTGAGCCATTTCAC

20/20

57.5/ 55.4

55.0/ 50.0

112

Nitric oxide biosynthetic genes

OsNIA1

LOC_Os08g36480

TCGGCAAGCACATCTTCGT

ACTTGGGGTGCTCGTTCTTG

19/ 20

54.9/ 57.5

52.6/ 55.0

138

OsNIA2

LOC_Os08g36500

TGTACCAGGTCATCCAGTCG

CGATGACGTACCACACCTTG

20 /20

57.5/ 57.5

55.0/ 55.0

162

Housekeeping gene

OsActin1

LOC_Os05g36290

CTAGCGGTCGAACAACTGGT

ACCGGAGGATAGCATGAGGA

20/ 20

57.5/ 57.5

55.0/ 55.0

102